# Fabrication and Polishing Performance of Diamond Self-Sharpening Gel Polishing Disk

**DOI:** 10.3390/mi15010056

**Published:** 2023-12-27

**Authors:** Lanxing Xu, Kaiping Feng, Liang Zhao, Binghai Lyu

**Affiliations:** 1College of Mechanical Engineering, Quzhou University, No. 78, North Jiuhua Road, Quzhou 324000, China; 221122020197@zjut.edu.cn (L.X.); 2112102214@zjut.edu.cn (L.Z.); 2China Ultra-Precision Machining Centre, Zhejiang University of Technology, No. 18, Chaowang Road, Hangzhou 310014, China; icewater7812@126.com

**Keywords:** gel polishing disk, self-sharpening, ultrafine diamond, 4H-SiC wafer, AlN powder

## Abstract

A diamond gel polishing disk with self-sharpening ability is proposed to solve the problem of glazing phenomenon in the gel polishing disks. Aluminum nitride (AlN) powder with silica sol film coating (A/S powder) is added to the polishing disk, and a specific solution is used to dissolve the A/S powder during polishing, forming a pore structure on the polishing disk. To realize the self-sharpening process, the dissolution property of the A/S powder is analyzed. The effect of A/S powder content on the friction and wear performance and the polishing performance of 4H-SiC wafers are investigated. Results showed that the friction coefficient of the polishing disk with 9 wt% A/S powder content is the most stable. The surface roughness *R_a_* of 2.25 nm can be achieved, and there is no obvious glazing phenomenon on the polishing disk after polishing. The surface roughness of the 4H-SiC wafer is reduced by 38.8% compared with that of the polishing disk with no A/S powder addition after rough polishing, and the 4H-SiC wafer then obtained a damage-free surface with a *R_a_* less than 0.4 nm after fine polishing by chemical mechanical polishing (CMP).

## 1. Introduction

As a third-generation semiconductor, 4H-SiC has the advantages of a large band gap (3.26 eV), high breakdown electric field strength (2.42 × 10^6^ eV), high thermal conductivity (4.8 W·cm^−1^·K^−1^), and high electron saturation drift rate (2.2 × 10^7^ cm·s^−1^), and its wafers have a wide range of applications in the manufacturing of high frequency, high power, high precision, and high temperature resistant electronic devices [1,2,3]. Due to the characteristics of high brittleness, high hardness, and high chemical inertness, the processing of 4H-SiC wafers is more difficult than that of traditional semiconductor materials [4]. Conventional polishing methods use resin bronze disks and diamond-free abrasive processing, but there is the problem of low processing efficiency. High-speed and high-pressure polishing can realize efficient processing of hard and brittle materials, but it is also easy to make free diamond abrasives splash around, resulting in serious waste [5,6]. Ultrafine diamond is an ideal abrasive for polishing hard and brittle materials. In order to reduce the loss of diamond abrasive and realize efficient precision machining, it is an effective measure to use diamond-fixed abrasive tools to polish 4H-SiC wafers [7,8].

The superfine abrasives in the fixed abrasive tools are prone to self-agglomeration, which affects the surface quality of the workpiece. Chandra et al. [9] found that the formation of scratches on the workpiece surface is related to the agglomeration of polishing powder. Egan et al. [10] found that fine-grained abrasives would agglomerate in the polishing slurry, resulting in micro-scratches on the surface of the workpiece. Li et al. [11] believe that self-agglomeration of ultrafine abrasives will lead to non-uniformity inside abrasive tools, thus causing mechanical damage to the workpiece surface. Compared with traditional dry hot pressing, wet mixing in the solution can make the ultrafine abrasives uniformly disperse in the resin solution by adding dispersants and coupling agents, etc., and reduce the mechanical damage caused by the self-agglomeration of abrasives [12]. In order to solve the problem of agglomeration of abrasives in hot pressing abrasive tools, Chen et al. [13] found that the ceramic bonded diamond grinding wheel prepared by the sol-gel method can improve the surface machining quality of PcBN tools. Wang et al. [14] found that compared with the grinding wheel prepared by the powder pressing method, the grinding wheel made by the sol-gel method has better surface consistency, no deep scratches, and the surface roughness reaches 0.049 μm. Luo et al. [15] prepared a new type of spherical polishing tool for processing seal stone by the sol-gel method, which obtained a smooth surface and was convenient for recycling with extremely low environmental pollution. Lu Jing et al. [16] found that after shear dispersion of ultrafine abrasives in an aqueous solution, particle agglomeration was effectively avoided and the stability was good. After processing the SiC wafer with a diamond polishing pad prepared by the sol-gel method, the ultra-smooth and non-destructive surface was obtained. Feng et al. [17] prepared a diamond gel polishing disk using polyvinyl alcohol (PVA) and phenolic resin (PF) as the binder to polish 4H-SiC wafers and obtained an ultra-smooth and non-destructive surface with a surface roughness of Ra less than 0.5 nm after CMP. Therefore, the use of the sol-gel method to prepare abrasive tools is an effective method to solve the agglomeration of ultrafine particles and also improve the surface quality of the workpiece.

With the growth of processing time, the abrasive grains on the gel abrasive tool will gradually passivate. Frictional heat and debris continue to accumulate and block the air holes leading to overheating glazing of the abrasive, which will affect the machining efficiency and the surface quality of the workpiece if it is not trimmed in time [18,19]. Ning et al. [20] found that the holding force of the bond on the particles in the grinding wheel was too high, which would make the passivation particles and debris unable to fall off, and further clog the chute, resulting in a sharp decrease in the cutting performance of the grinding wheel. Gourhari et al. [21] concluded that the passivation of abrasive grains leads to a drastic reduction in machining accuracy and effectiveness. In order to solve this problem, improving the self-sharpening of abrasive tools is an effective way [22]. Wang et al. [23] developed an abrasive water jet system to dress fixed abrasive lapping pads, which significantly improved the machining efficiency of lapping pads. Guan et al. [24] suggested that an appropriate amount of porosity in the abrasive tool can promote the self-sharpening performance of the abrasive tool. Li et al. [25] prepared a porous diamond abrasive tool with high self-sharpening properties by thermos-chemical etching. Compared with ordinary diamond abrasive tools, the grinding efficiency was increased by 29.6%, and the surface roughness of the workpiece was reduced by 27.5%. In order to avoid a glazing phenomenon of the fixed abrasive pad (FAP) after long time processing, Chen et al. [26,27] developed a new method of loose abrasive assisted fixed agglomerated diamond abrasive grinding (LA-FADAL). The self-sharpening ability of FADAP is enhanced by the scouring and rolling of loose particles. Therefore, improving the self-sharpening property of the abrasive tool can play a role in increasing the machining efficiency and improving the surface quality of the workpiece.

In order to solve the problems of the surface glazing phenomenon and abrasive passivation of gel polishing disks after a long polishing time, the self-sharpening performance of polishing disks is improved by adding water-resistant AlN powder as a “soluble” filler inside the disk. In this paper, the dissolution property of A/S powder was first analyzed. Then, friction and wear experiments and polishing experiments were carried out to investigate the effect of A/S powder content on the friction and wear performance of polishing disks and the surface quality of 4H-SiC wafers.

## 2. Self-Sharpening Mechanism and Preparation of a Polishing Disk

### 2.1. Self-Sharpening Mechanism

In order to solve the glazing phenomenon on the surface of gel polishing disks after a long polishing time, water-resistant AlN powder is added inside the polishing disk. NaOH solution is added dropwise to the polishing disk during processing. Since AlN is easily hydrolyzed and the preparation of the gel polishing disk is a wet mixing process, a layer of hydrophobic silica sol film is coated on the surface of AlN to avoid AlN hydrolysis during the preparation of polishing disks. As shown in Figure 1, when a polishing disk polishes a 4H-SiC wafer, the diamond grains are gradually passivated as the polishing time increases. If there are drops of NaOH solution in the processing area of the polishing disk, the solution will destroy the silica sol in the surface layer of A/S powder and at the same time promote the hydrolysis of AlN inside A/S. As a result, Equations (1)–(3) show that the A/S powder on the surface of the polishing disk will quickly dissolve in the solution [28], forming new pores. The chip space and heat dissipation capacity of the polishing disk are improved.
(1)$$2NaOH+{SiO}_{2}={Na}_{2}{SiO}_{3}+H_{2}O$$
(2)$${AlN+4H}_{2}O\underset{\left(NaOH\right)}{\to }{Al(OH)}_{3}+{NH}_{4}OH$$
(3)$$NaOH+{Al(OH)}_{3}\to {NaAlO}_{2}+2H_{2}O$$

### 2.2. Preparation of Polishing Disk

#### 2.2.1. Preparation of A/S Powder

In order to avoid the hydrolysis of AlN during the preparation of polishing disks, the hydrolysis reaction of ethyl orthosilicate (TEOS) (Equation (4)) is used to prepare a silica sol, which is then cloaked on the surface of the AlN powder to obtain water-resistant AlN powder. As in Figure 2, the A/S preparation process is as follows: (1) prepare acetic acid solution, acetone solution, and TEOS solution; (2) take AlN powder, add acetone solution into the TEOS solution sequentially, and stir well; (3) continue to add acetic acid solution and a small amount of deionized water; (4) put in a 50 degree water bath environment static for 30 min; (5) dry and mash the formation of the gel in a 325 mesh sieve; and (6) carry out heat treatment in a 700-degree oven.
(4)$${Si({OCH}_{2}{CH}_{3})}_{4}+2H_{2}O\to {SiO}_{2}+4C_{2}H_{5}OH$$

#### 2.2.2. Preparation of the Polishing Disk

The composition of the gel polishing disk is shown in Table 1, and fillers are usually added to resin-bonded abrasive tools to improve their mechanical and polishing properties. The strong thermal conductivity of copper powder can reduce the local overheating phenomenon in the processing area and enhance the abrasion resistance of the polishing disks; ZnO powder can significantly improve the antifriction and wear resistance of the polishing disk; the wetting agent and the toughening agent can improve the wettability of the glue on the particles and the toughness of the embryo. The polishing disk preparation process is shown in Figure 3: (1) Disperse diamond micro-powder, A/S powder, filler, etc., in water to obtain a suspension. (2) Configure polyvinyl alcohol and phenolic resin mass ratio of 1:5 co-mingled glue, add it to the suspension, and mix to get the slurry. (3) When the slurry is well mixed, sieve it, pour it into a circular mold, and place it in a cycle of freezing at −20 °C for 5 times. (4) When the embryo is thawed, place it in a constant temperature and humidity oven at a temperature of 40 °C and a humidity of 20% to dry for 100 h, and, finally, place it in an oven at 180 °C for sintering and curing.

## 3. Equipment and Experimental Set-Up

### 3.1. Equipment

A Shanghai Lei magnetic PHS-3C acidimeter was used to measure the pH value of the solution (precision 0.01); a DF-1 type collector-type magnetic stirrer was used for water bath heating; the RTS series laser confocal microscopic Raman spectrometer was used to measure the products of A/S powder dissolution; Hitachi SU8010 SEM was used to observe the surface micro-morphology of the polishing disks, Shimadzu DUH-211 hardness tester was used to measure the hardness of the polishing disks; the bending strength of the polishing disk was measured by the tensile testing machine; the mass change of the 4H-SiC wafer was weighed by Li-Chen FA124C electronic analytical balance; the friction coefficient of the polishing disk was measured by HT1000 high temperature friction and wear testing machine of Zhongkekai Technology Co., Ltd., Lanzhou, China. (the friction pair used in the tester is a silicon nitride ceramic ball with a diameter of 8 mm); keyence digital microscope was used to observe the size of the abrasion marks on the polishing disk specimen; KLA TencorMicroXAM 1200 white light interferometer was used to measure the microscopic morphology and surface roughness of the 4H-SiC wafer after polishing.

### 3.2. Setup of Experimental Set-Up

#### 3.2.1. Setup of Dissolution Experiment

The content of AlN in A/S powder is determined by distillation separation-neutralization titration [29]. The experimental device is shown in Figure 4. The determination principle is that A/S powder can react with alkali to produce ammonia gas, and the resulting ammonia gas is collected by boric acid solution. Then, the AlN content in A/S powder is calculated by hydrochloric acid titration [30]. Firstly, 150 mL of NaOH solution of 40 wt% concentration is added to a conical flask and heated to 90 degrees by a water bath. A g of A/S powder is weighed into the conical flask and the A/S powder is allowed to fully react with the NaOH solution for 12 h. The released ammonia passes through the condensing tube and is absorbed by a 2 wt% boric acid solution. Then 100 mL of the reacted boric acid solution is taken into a beaker and a few drops of methyl red-methylene blue is added as an indicator. A hydrochloric acid solution of concentration C is used to titrate the boric acid solution, and the titration is completed when the solution changes from a green color to a purplish red color. The consumption of hydrochloric acid is recorded as V1. The consumption of hydrochloric acid is recorded as V2 by using an equal volume of deionized water for the blank experiment. The content of AlN in the A/S powder can be calculated using Equation (5).

Several groups of A g A/S powder are placed in different concentrations of NaOH solution, and the dissolution tests are carried out under the magnetic stirring constant temperature water bath device at 40 degrees. After 60 min, the slurry is taken out and the filter residue is filtered out, and the filter residue is heated in a drying oven at 100 °C for 5 h. The content of AlN in filter slag is determined again by distillation separation-neutralization titration. The removal rate of AlN is determined by Equation (6). All dissolution tests are repeated 3 times and averaged. The dissolution degree of A/S powder is indirectly reflected by the removal rate of AlN.
(5)$$\omega_{AlN}=\frac{0.041C_{HCl}(V_{1}-V_{2})}{A}$$
(6)$$X=\frac{\omega_{AlN}-\omega_{1}}{\omega_{AlN}}$$
where $\;\omega_{AlN}$ is the content of AlN in A/S powder (wt%); $C_{HCl}$ is the concentration of hydrochloric acid ($mol\cdot L^{-1}$); $V_{1}$, $V_{2}$ is the amount of hydrochloric acid consumed (mL); *A* is the mass of A/S powder (g); $\omega_{1}$ is the content of AlN in the filtrate residue (wt%); and *X* is the AlN removal rate.

#### 3.2.2. Setup of the Polishing Experiment

The polishing machine adopts a Shenyang Kejing UNIPOL-1000S plane polishing machine, and the polishing object is a 4H-SiC wafer with a diameter of 100 mm. The diameter of the polishing disk is 280 mm, and the working pressure is 0.2 MPa. The rotational speed of the polishing disk and the workpiece is 120 r/min and 30 r/min, respectively. The NaOH solution is added to the polishing disk during polishing. The 4H-SiC wafers are the precision ground before polishing, and the average surface roughness Ra is 71 nm. As shown in Figure 5, diamond gel polishing disks are used to rough-polish the 4H-SiC wafers. The effects of different A/S contents on the surface quality of 4H-SiC wafers and polishing disks are analyzed. Polyurethane polishing pads and silica sol polishing solution are selected to conduct the CMP fine polishing. The material removal rate of the polishing disks is measured by the mass change of the 4H-SiC wafers (Equation (7)). Surface roughness selects the 10 measurement points in Figure 5, and the average value is taken after measurement.
(7)$$v_{MRR}=\frac{∆m}{\rho ts}$$
where  ∆m is the mass change of a 4H-SiC wafer before and after polishing, ρ is the density of 4H-SiC wafer ($\rho =3.2\;g/{cm}^{3}$), t is the polishing time, and s is the bottom area of the 4H-SiC wafer.

## 4. Results and Discussion

### 4.1. Dissolution of A/S Powder

During the polishing process of the polishing disk, NaOH aqueous solution and mechanical friction will destroy part of the silica sol layer on the A/S surface. The AlN inside the A/S powder was exposed to water for the hydrolysis of AlN. The speed of the hydrolysis of AlN had a great influence on the dissolution speed of A/S powder, so speeding up the hydrolysis speed of AlN is conducive to the rapid dissolution of A/S powder. If the A/S powder can not be dissolved in time to evacuate the heat of the disk surface, some areas of the disk surface may be overheated and glazed, hindering the further processing of abrasive tools on the workpiece, affecting the processing efficiency. Therefore, the hydrolytic properties of AlN powder within A/S powder were investigated before further studying the dissolution process of the overall A/S powder.

AlN is a type III-V strongly covalently bonded nitride with high thermal conductivity, high strength, and low dielectric constant, which is widely used in microelectronics, automation, semiconductors, and other fields [31]. However, there is an induction period before the hydrolysis reaction of AlN occurs at an ambient temperature below 90 degrees Celsius, during which the hydrolysis reaction of AlN is greatly inhibited [32]. Figure 6a shows the hydrolysis pH value changes of AlN (3 wt% AlN content) at different particle sizes. Figure 6b shows the hydrolysis pH value changes of AlN (3 wt% AlN content, 1 μm particle size) at different temperatures. As shown in Figure 6, the induction period of AlN is extremely long at room temperature. Reducing the particle size of the AlN powder and increasing the temperature of the water bath can reduce the time required for induction. In order to further reduce the induction period time of AlN, AlN powder is immersed in phosphoric acid solution (pH: 2.5), deionized water (pH: 6), and NaOH solution (pH: 10) at room temperature to explore the hydrolysis of AlN under an acid-base environment. The content of A/S in the solutions is 3 wt%, and the pH of the solutions is tested by a pH tester every 5 min. As can be seen in Figure 7, there is no significant change in pH when the initial pH is 2.5. It is presumed that the low concentration of OH^-^ inhibited the hydrolysis of AlN, while the phosphoric acid solution generated a protective film on the surface of AlN through adsorption and hydrogen bonding, hindering the contact between the powder and water molecules [33]. At an initial pH of 6, there is no significant change in pH within 60 min. This was due to the fact that an amorphous aluminum hydroxide gel was formed during the induction period, which covered the surface of the AlN powder and prevented the hydrolysis reaction of AlN in contact with water molecules [34]. When the initial pH is raised to 10, the pH value gradually increases with time. It reaches the maximum value at 40 min, and the pH is basically unchanged. This is due to the fact that as the pH increases, the stability of amorphous aluminum hydroxide gel decreases, and the solubility increases greatly. There were a large number of ${OH}^{-}$ ions in the solution, and the highly electronegative hydroxyl group would destroy the Al-N bond and form the ${Al(OH)\;}_{4}^{-}$ substance required for the growth of thin hydrotalcite [35]. The time of the induction period is greatly shortened, and the pH value gradually rises. Therefore, increasing the ambient temperature and adding NaOH can effectively promote the hydrolysis of AlN.

An amount of 3 g of A/S powder is tested by XRD. Another 3 g of A/S powder is added to 100 mL of water solution. After 100 h, the powder is filtered out and dried for an XRD test. The results are shown in Figure 8, the A/S powder has good water resistance and is not easy to hydrolyze in wet mixing. The surface layer of the powder had a dense silica sol coating (Figure 9), and no obvious new phases were generated in the XRD pattern. The content of AlN in the A/S powder was calculated to be 90.6% by Equation (5), so the dissolution degree of A/S was indirectly reflected by the rate of AlN removal. Several groups of 20 g A/S powder are weighed and placed them different concentrations of NaOH solution. The liquid–solid ratio was 15 mL/g, the water bath temperature was 40 degrees (similar to the temperature of the polishing disk processing area), and the reaction time was 60 min. As shown in Figure 10, the NaOH was able to promote the dissolution of A/S. With the increase in NaOH concentration, the AlN removal rate continued to increase, and the dissolution degree of the powder gradually deepened. When the NaOH concentration reached 10 wt%, the AlN removal rate in the A/S powder reached 61.7%, which basically enabled part of the A/S powder to be dissolved in the solution. In conjunction with the literature [36], the NaOH concentration was set to 10 wt%. Three sets of solutions after the reaction of 10 wt% NaOH with A/S powder were taken for Raman spectroscopy. The solution in Figure 11 is able to detect the presence of significant aluminate ions ${Al(OH)}_{4}^{-}$ (Raman shifts: 621 cm^−1^ and 320 cm^−1^) and a small amount of silicon oxide ions (Raman shifts: 445 cm^−1^) [37]. It is speculated that sodium meta-aluminate and sodium silicate produced by A/S dissolution are soluble in water. Reaction products are easy to rinse and not easily retained on the surface of the polishing disk. To sum up, it can be seen that the alkaline environment is favorable to promote the rapid dissolution of A/S powder. In order to effectively play the role of A/S powder in polishing disks, a small amount of a 10 wt% NaOH aqueous solution was added during processing, so that the A/S powder on the surface of the polishing disks could be quickly dissolved (Equations (1)–(3)) to produce a new pore structure, which served to enhance the chip space and heat dissipation capacity and facilitated the self-sharpening of the polishing disks.

### 4.2. Effect of A/S Particle Size on the Performance of Polishing Disks

The particle size of A/S itself is difficult to control, so the particle size of A/S powder was changed indirectly by changing the particle size of AlN inside A/S to study the effect of particle size on the polishing disk. Three kinds of A/S powders with different particle sizes (AlN particle size inside A/S, group A: 0.2 μm; group B: 1 μm; group; C: 5 μm) were added into the polishing disk (A/S powder content of 9 wt%). The 4H-SiC wafer was polished by placing three kinds of polishing disks on the plane polishing machine; 10 wt% NaOH aqueous solution was added during polishing, and the polishing time was 60 min. The mechanical properties of the polishing disks were tested after the completion of the process. Figure 12 shows the viscosity of slurry before polishing disc molding. Figure 13 shows the surface morphology of polishing disk specimens before and after polishing. Figure 14 shows the mechanical properties of three kinds of polishing disks after polishing. From Figure 12 and Figure 14, it can be seen that the flexural strength and hardness of the polishing disks show an overall decreasing trend as the particle size of the A/S powder increases. It is speculated that with the increase in A/S particle size, the viscosity of the slurry increases, and the uniformity inside the polishing disk decreases. At the same time, the surface layer of A/S powder on the polishing disk is dissolved, the porosity increases, and the mechanical properties of the surface of the polishing disk decrease. After the polishing disk of group A is processed, the glazing layer shown in Figure 13a appeared on the surface of some areas of the polishing disk, covering the surface of the abrasive grains and affecting the polishing effect of the abrasive tools on the workpieces. In contrast, there is no obvious glazing phenomenon in group B and C. As the polishing disk is prepared by wet mixing, the particle size of A/S powder increases, and the viscosity of the slurry increases, which will reduce the internal bonding tightness of the polishing disk. The surface of the sintered polishing disk is prone to microcracks as shown in Figure 13c. The large pores generated by the dissolution of the large particle size A/S powder will greatly reduce the holding capacity of the binder, and the flexural strength and hardness of the abrasive tool will be reduced consequently. The polishing disk is prone to brittle fracture during processing as shown in Figure 13c. In addition, abrasive grains will fall off prematurely, and the accumulation of a large number of abrasive particles and debris will affect the polishing effect of the polishing disk. As shown in Figure 13b, the surface of group B polishing disc after polishing has no obvious glaze phenomenon, and its strength and hardness are higher than those of group C. In summary, the particle size of AlN inside the A/S powder is set to 1 μm in this paper according to the considerations of mechanical properties and polishing stability.

### 4.3. Friction Wear Test

Diamond gel polishing disk specimens mixed with 0, 3, 6, 9, 12, and 15 wt% AlN were prepared according to the fabrication method of Figure 5, and the specimens were tested in friction wear experiments. The polishing time was 60 min, and 10 wt% NaOH solution was added dropwise on the specimen surface during the test. The polishing disk samples with different A/S powder contents were produced, and the apparent porosity of the polishing disk specimens was calculated by weighing the dry weight, wet weight, and floating weight based on Archimedes’ principle [38]. The initial apparent porosity of the different samples was first measured. Then, the polishing disk specimens were immersed in NaOH solution for 1 h. Some of the A/S powder in the specimens was dissolved, and the apparent porosity was measured again after removing and drying the specimens. As shown in Figure 15, when the polishing disc sample was soaked in NaOH solution (time: 1 h), the A/S inside it was dissolved in the solution by NaOH, and the pores increased. With the increase in A/S content, the porosity of polishing disk also increased. When the number of pores is too much, they will interact with each other, leading to collapse of some of the pores; therefore, the growth of apparent porosity decreases slowly at higher A/S powder contents. Figure 16 shows the instantaneous coefficient of friction with time for different polishing disks. Figure 17 shows the friction ring width of the polishing disk specimens after 60 min. Figure 18 shows the hardness of the polishing disk specimens with different A/S contents at the end of the friction wear test. As can be seen from Figure 16 and Figure 18, the average coefficient of friction of the polishing disk shows an overall increasing trend with the rise of the A/S powder content. Polyvinyl alcohol (PVA) is a thermoplastic material; when the A/S is 0 and 3 wt%, polyvinyl alcohol on the surface of the polishing disk is easy to overheat and soften and decompose, forming a glazed film over the surface of the polishing disk. The bonding strength of the bonding point decreases, the intermolecular attraction decreases, and the phenomenon of slipping occurs. The performance is characterized by a large degree of instantaneous friction coefficient fluctuations. When the A/S is 6 and 9 wt%, the increase in porosity enhances the chip space and heat dissipation ability of the polishing disk. The thermal recession temperature of polyvinyl alcohol and phenolic resin is effectively increased, and the glazing phenomenon is alleviated. The instantaneous friction coefficient is relatively stable. When the A/S powder content continues to increase, the instantaneous friction coefficient fluctuation changes in a large magnitude and poor stability. This is because the A/S powder content Is too high, the surface is more porous, and the ability of the bond to hold the abrasive grains is reduced. Abrasive grains are easily shed and pressed into the surface of the polishing disk or the formation of dislodged pits, affecting the stability of the test. At the same time, the surface hardness of the polishing disk decreases dramatically. When the silicon nitride ball is in contact with the polishing disk for testing, the surface of the polishing disk is prone to brittle rupture, resulting in poor stability of the instantaneous coefficient of friction and large fluctuations.

The abrasion marks of the silicon nitride balls on the surface of the polishing disk specimen in the friction wear test are approximated as sphericity, so the wear volume can be approximated by the depth of the abrasion marks, and the derivation of the equations is shown in Equations (8) and (9) [17]. From Figure 17, it can be seen that the widths of the friction ring are 3009 μm, 3268 μm, 3382 μm, 3527 μm, 4113 μm, and 4688 μm (corresponding to the A/S powder content of 0, 3, 6, 9, 12, and 15 wt%, in that order). The wear volumes of each polishing disk specimen are calculated as 224 mm^3^, 292 mm^3^, 418 mm^3^, 524 mm^3^, 710 mm^3^, and 931 mm^3^, based on the formula. It can be seen that as the A/S powder content rises, the wear volume of the polishing disk also rises. When the A/S powder content reaches above 9 wt%, the wear volume of the polishing disk starts to increase sharply. It is presumed that this is because the porosity is too high, which greatly reduces the ability of the bond to hold the abrasive grains. The abrasive grains fall off faster, and the abrasive grains scattered on the surface of the polishing disk can easily cause scratches, white spots, and other damage to the polished surface. At the same time, the increase in porosity makes the surface hardness of the polishing disk decrease; silicon nitride and the polishing disk surface contact are prone to brittle rupture, which leads to a rapid increase in the tool wear phenomenon. Therefore, the content of A/S powder should not be too high.
(8)$$V=\left(r^{2}\arccos\left(\frac{r-h}{r}\right)-\left(r-h\right)\sqrt{2rh-h^{2}}\right)\times l$$
(9)$$h=r-\sqrt{r^{2}-{\left(\frac{a}{2}\right)}^{2}}$$
where V is the wear volume (mm^3^); r is the radius of the silicon nitride ball (mm); l is the length of the abrasion mark (mm); h is the depth of the abrasion mark (mm); and a is the width of the friction ring (mm).

### 4.4. Polishing Experiment

Taking A/S as a variable, polishing disks with A/S powder contents of 0, 3, 6, 9, 12, and 15 wt% were taken for rough polishing comparison experiments. The polishing objects were 4H-SiC wafers, the polishing time was 60 min, and 10 wt% NaOH aqueous solution was added dropwise during the process. Figure 19 shows the surface morphology of 4H-SiC wafers after rough polishing with six types of polishing disks. As shown in Figure 19, the polishing of SiC by a polishing disc is a process from brittle removal to plastic removal in which the brittle process is dominant. It is easy to cause surface damage in the process of brittleness removal [39,40]. Figure 20 shows the surface roughness and material removal rate of 4H-SiC wafers after polishing. Figure 21 shows the surface topography of the polishing disk after rough polishing. From Figure 19, Figure 20 and Figure 21, it can be seen that the material removal rate increases and then decreases with the increase in the A/S powder content, and the surface roughness of 4H-SiC wafers varies between 2.25 nm and 4.65 nm. When the A/S powder content is 0 wt%, the material removal rate is the lowest, and the surface of the abrasive tool is easily covered by a glazed layer as shown in Figure 21a. The contact area between the abrasive grains and the workpiece is reduced, the polishing effect is poor, and the surface roughness is high in Figure 19a. When the A/S powder content is increased to 9 wt%, the A/S powder on the polishing disk is dissolved, providing a larger chip holding space. The holding power of the polishing disk on the abrasive grains is reduced, the passivated abrasive grains fall off, the new abrasive grains are exposed, and the self-sharpening ability of the polishing disk is improved. The highest material removal rate of 1.49 μm/h is achieved with the best machining results. The surface of the workpiece has few machining marks, and the surface roughness is as low as 2.25 nm in Figure 19d. When the A/S powder content continues to increase, the number of pores produced by the dissolution of A/S powder increases and the hardness of the polishing disk decreases. Some areas of the polishing disk are prone to brittle rupture as shown in Figure 21e,f. At the same time, the ability of the bond to hold the abrasive grains continues to decrease, the abrasive grains may be dislodged prematurely, and debris and dislodged abrasive grains accumulate in large quantities on the polishing disk, which can easily leave deep scratches on the workpiece, as shown in Figure 19e,f. The surface roughness of the polishing disk also deteriorates, and the processing stability deteriorates. The surface roughness of the workpiece increases continuously. The surface roughness of 4H-SiC wafers after polishing with a polishing disk of 15 wt% A/S powder content is the worst, reaching 4.65 nm, as shown in Figure 19f.

Compared with the ordinary diamond gel polishing disk, the addition of A/S powder effectively promotes the self-sharpening effect of the polishing disk, alleviates the glazing phenomenon, and improves the surface quality of the workpiece. In conclusion, the surface quality of the workpiece polished with a 9 wt% gel polishing disk is the best, and the surface roughness of the 4H-SiC wafer workpiece after rough polishing is as low as 2.25 nm. The 4H-SiC wafer continues to be processed by CMP finishing polishing, and an ultra-smooth and non-destructive surface with a roughness of 0.39 nm is obtained, as shown in Figure 22.

## 5. Conclusions

The aim of this study is to develop a gel polishing disk with self-sharpening effect, which is promoted by adding water-resistant AlN powder to the polishing disk. The main findings of this research work are as follows:(1)The larger the AlN particle size inside the A/S powder, the longer the induction period required to start hydrolysis. Increasing the water bath temperature and adding NaOH can effectively shorten the induction period of AlN. The NaOH solution is added to the polishing disk, and the A/S powder dissolves to form new pores to increase the chip space and enhance self-sharpness.(2)With the increase in A/S powder content, the average coefficient of friction gradually increases, and the wear of the polishing disk becomes more serious. The instantaneous friction coefficient of the polishing disk is most stable and the wear is moderate when the A/S powder content is 9 wt%.(3)The polishing disks with different A/S contents are prepared to rough-polish the 4H-SiC wafer workpieces. With the increase in A/S powder content, the surface roughness of the workpieces decreased first and then increased, and the material removal increased first and then decreased. The best 4H-SiC wafer surface is polished by a 9 wt% A/S powder content gel polishing disk, with a surface roughness of 2.25 nm and fewer surface scratches. The polishing disk had a strong self-sharpening effect, and there is no obvious glazed layer on the surface to hinder processing. Ultra-smooth 4H-SiC wafer surfaces with a Ra less than 0.4 nm are obtained after CMP fine polishing.

## Figures and Tables

**Figure 1 micromachines-15-00056-f001:**
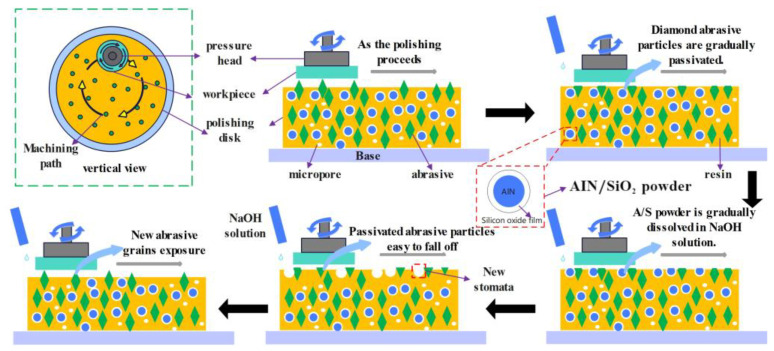
Promoting process of A/S powder on the self-sharpening of the polishing disk.

**Figure 2 micromachines-15-00056-f002:**
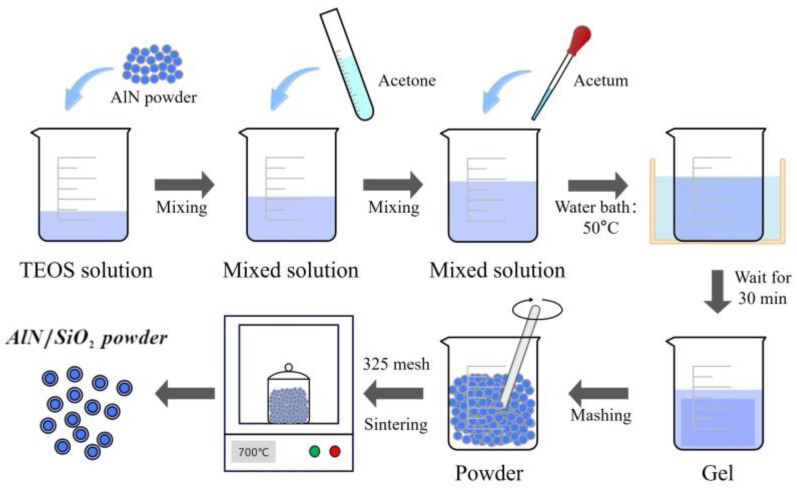
Preparation process of A/S powder.

**Figure 3 micromachines-15-00056-f003:**
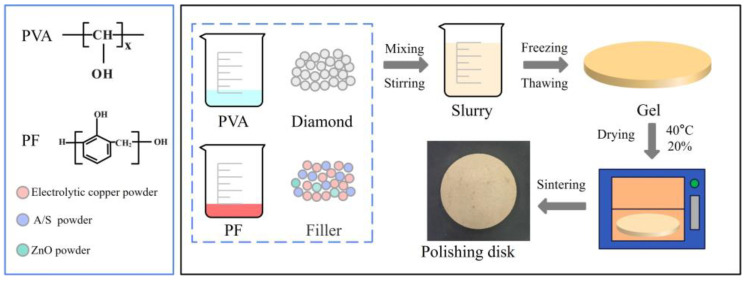
Preparation of the gel polishing disk.

**Figure 4 micromachines-15-00056-f004:**
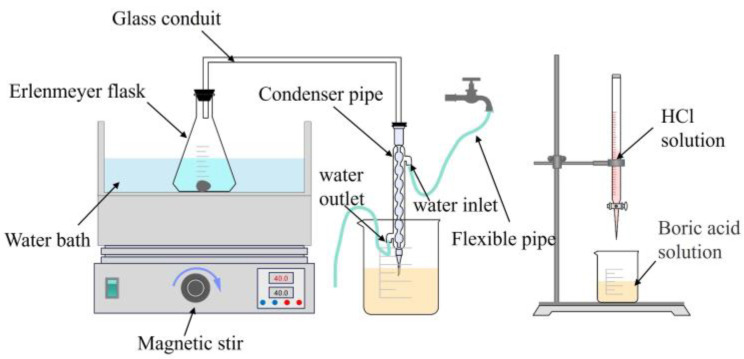
AlN content test device.

**Figure 5 micromachines-15-00056-f005:**
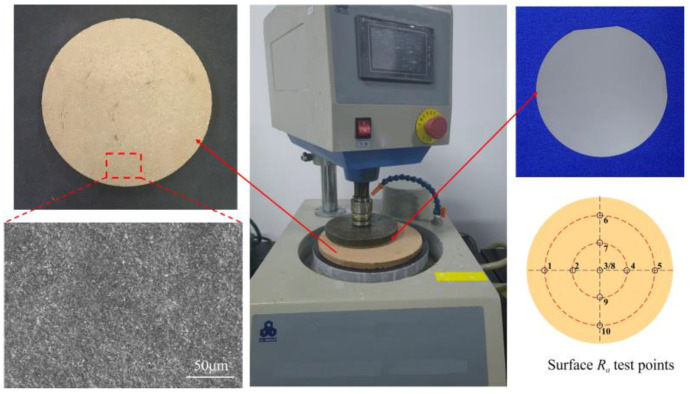
Polishing device.

**Figure 6 micromachines-15-00056-f006:**
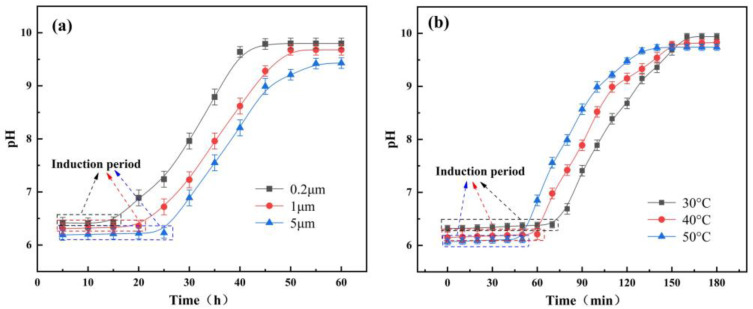
Hydrolysis test of AlN: (**a**) influence of different powder particle sizes on the pH of the AlN solution at room temperature 23 degrees; (**b**) influence of different water bath temperatures on the pH of the AlN solution.

**Figure 7 micromachines-15-00056-f007:**
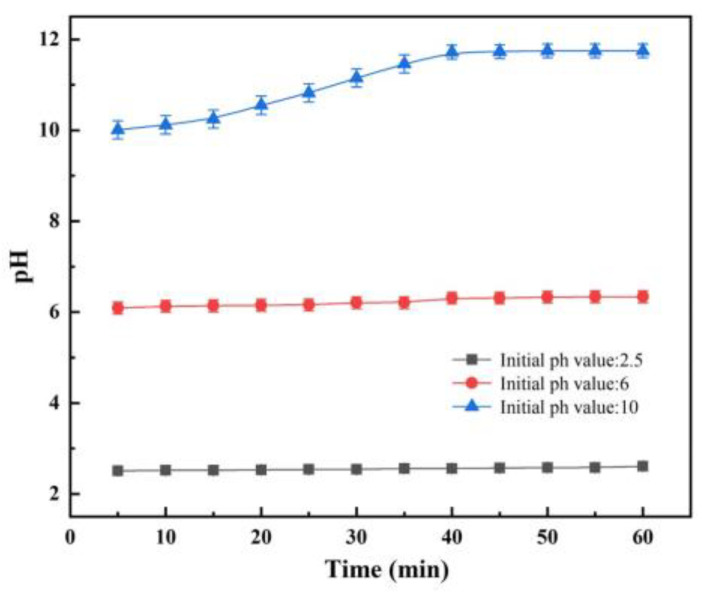
Effect of the initial pH value on the hydrolysis of AlN.

**Figure 8 micromachines-15-00056-f008:**
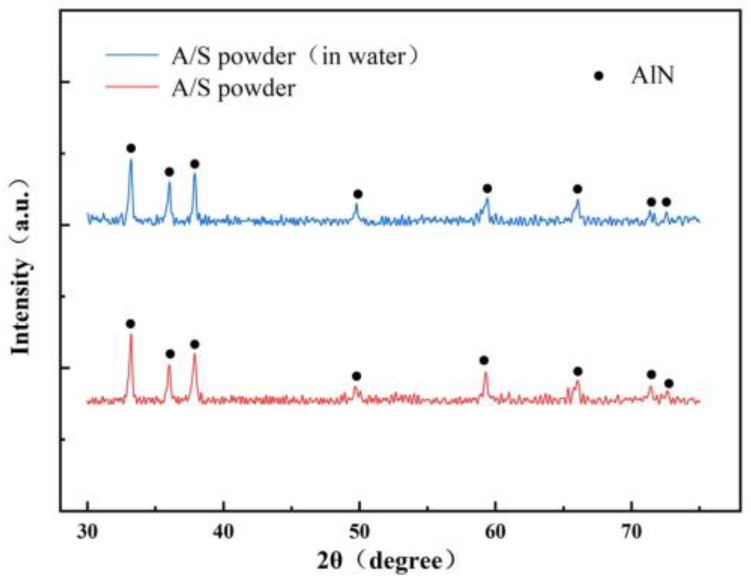
Water resistance test of A/S powder.

**Figure 9 micromachines-15-00056-f009:**
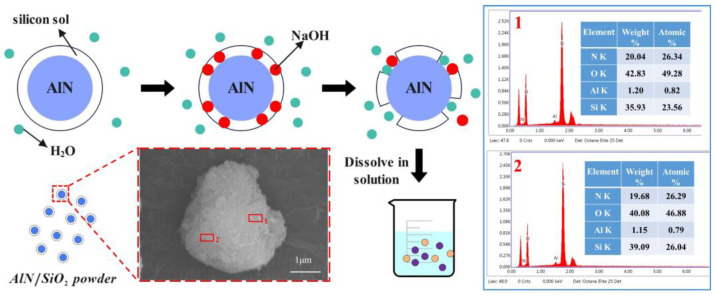
Schematic diagram of the dissolution process and EDS analysis for A/S powder.

**Figure 10 micromachines-15-00056-f010:**
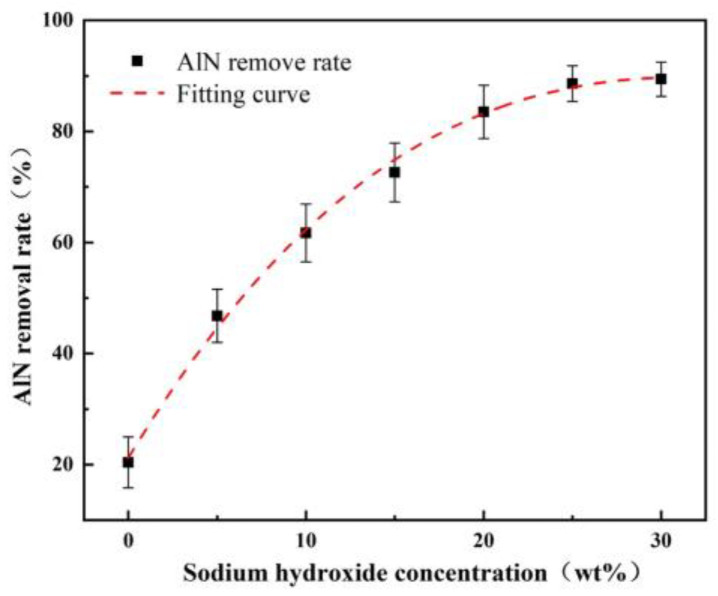
Effect of NaOH concentration on the removal rate of AlN in A/S powder.

**Figure 11 micromachines-15-00056-f011:**
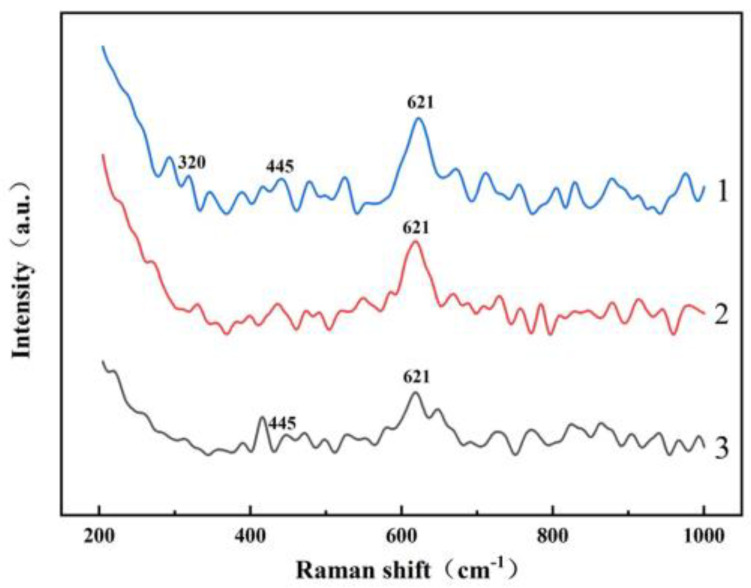
Raman spectra of reaction products (Measurement group:1, 2, 3).

**Figure 12 micromachines-15-00056-f012:**
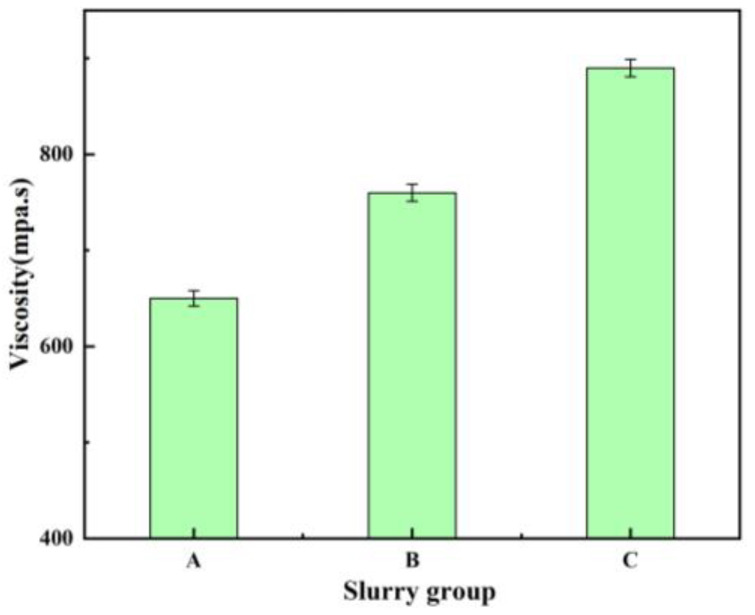
Effect of different AlN particle sizes on slurry viscosity.

**Figure 13 micromachines-15-00056-f013:**
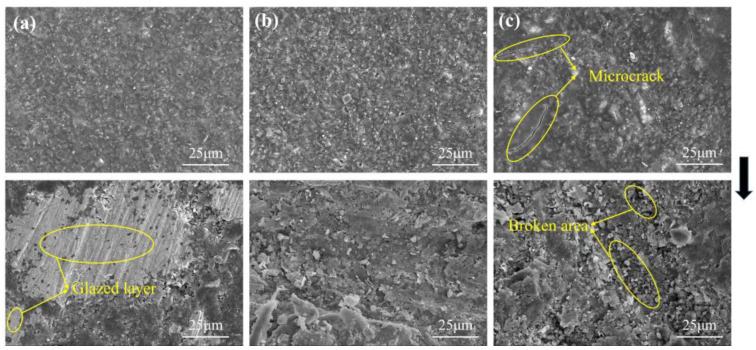
The surface morphology of the polishing disk before and after polishing: (**a**) group A; (**b**) group B; (**c**) group C.

**Figure 14 micromachines-15-00056-f014:**
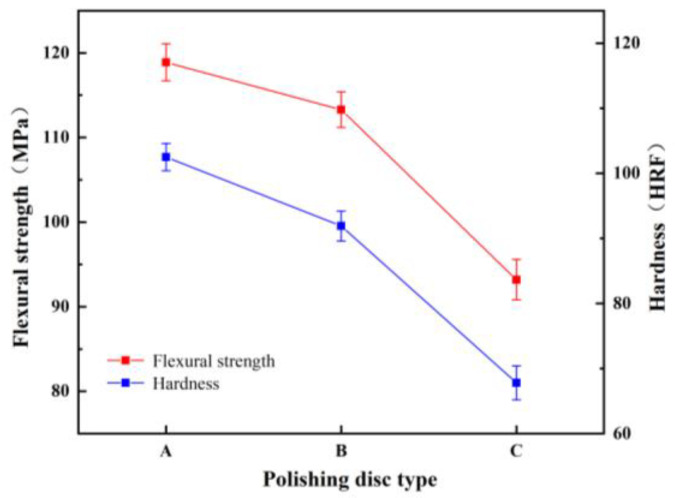
The mechanical properties of the polishing disks after polishing.

**Figure 15 micromachines-15-00056-f015:**
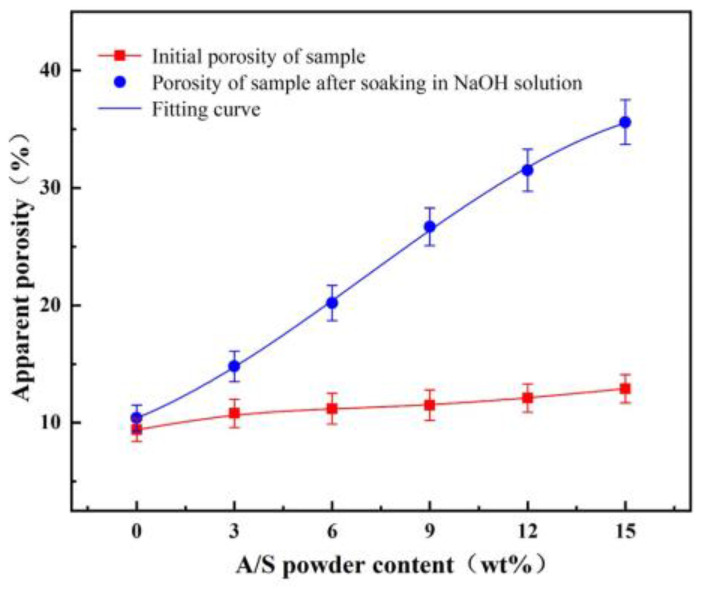
The apparent porosity of the polishing disk specimens.

**Figure 16 micromachines-15-00056-f016:**
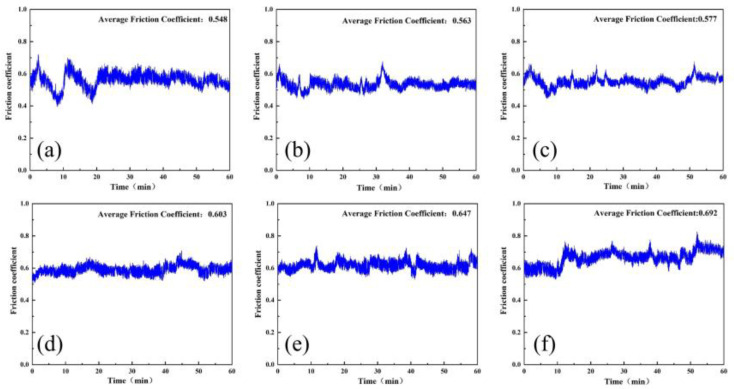
Instantaneous friction coefficient of the polishing disk specimens with different A/S powder contents: (**a**) 0wt%; (**b**) 3wt%; (**c**) 6wt%; (**d**) 9wt%; (**e**) 12wt%; (**f**) 15wt%.

**Figure 17 micromachines-15-00056-f017:**
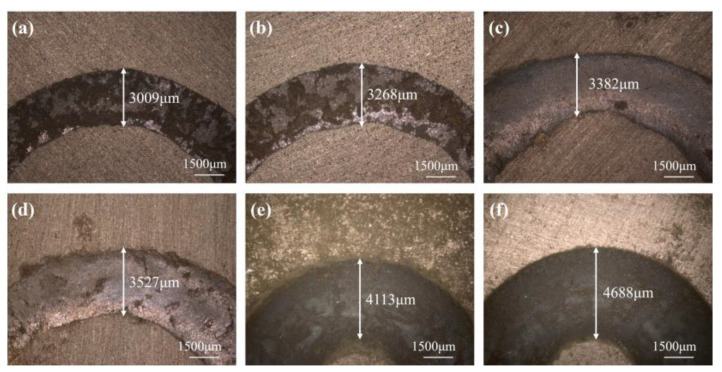
Surface morphology of friction and wear on the polishing disk specimens with different A/S powder contents: (**a**) 0 wt%; (**b**) 3 wt%; (**c**) 6 wt%; (**d**) 9 wt%; (**e**) 12 wt%; (**f**) 15 wt%.

**Figure 18 micromachines-15-00056-f018:**
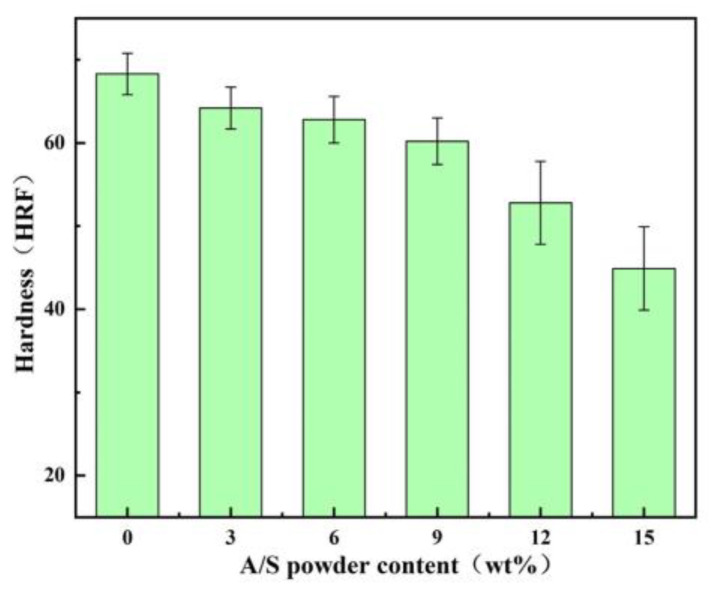
Hardness of the polishing disk specimens with different A/S powder contents after rough polishing.

**Figure 19 micromachines-15-00056-f019:**
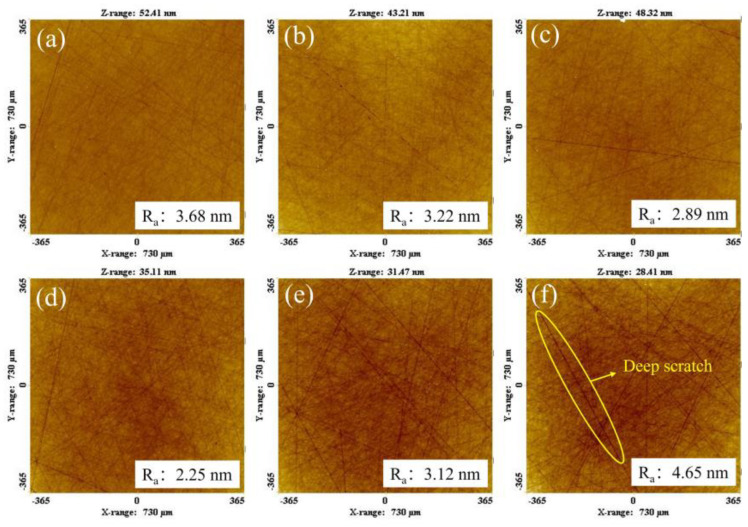
Effect of the A/S powder content on the surface quality of 4H-SiC wafers after rough polishing: (**a**) 0 wt%; (**b**) 3 wt%; (**c**) 6 wt%; (**d**) 9 wt%; (**e**) 12 wt%; (**f**) 15 wt%.

**Figure 20 micromachines-15-00056-f020:**
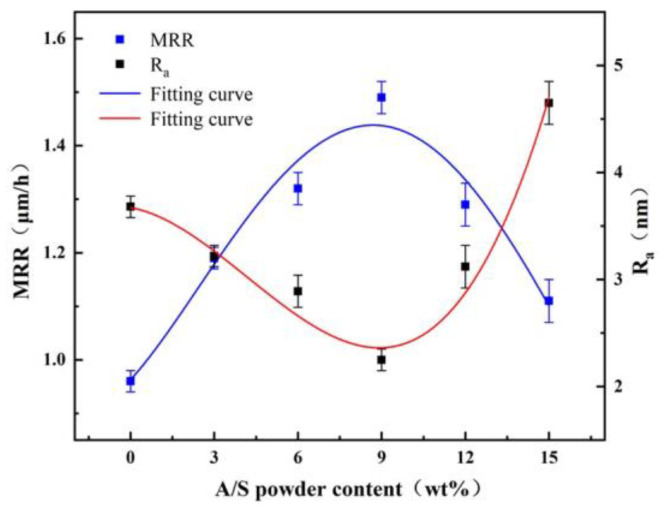
Material removal rate and surface roughness of 4H-SiC wafers after rough polishing.

**Figure 21 micromachines-15-00056-f021:**
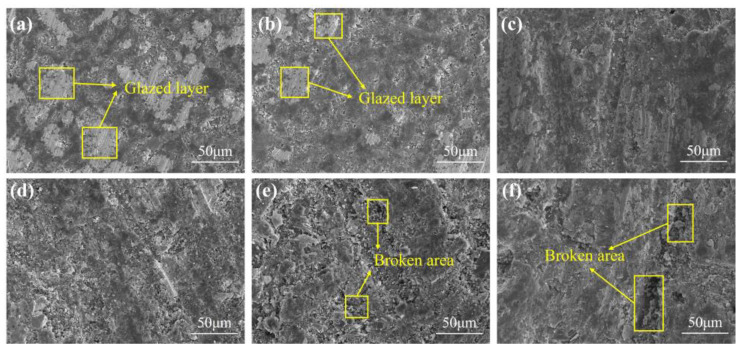
Surface morphology of polished disks after rough polishing: (**a**) 0 wt%; (**b**) 3 wt%; (**c**) 6 wt%; (**d**) 9 wt%; (**e**) 12 wt%; (**f**) 15 wt%.

**Figure 22 micromachines-15-00056-f022:**
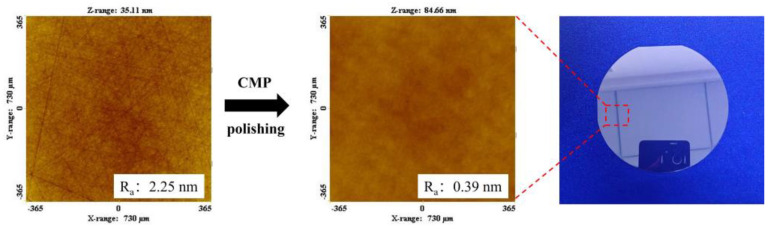
The 4H-SiC wafer after CMP finishing polishing.

**Table 1 micromachines-15-00056-t001:** Main components of the gel polishing disk.

Component	Granularity/μm	Solid Content/wt%
PVA/PF resin		15
Diamond powder	2.5	20
A/S powder		9
Electrolytic copper	3	40
ZnO powder	1	5
Graphite powder	2	3
Wetting agent		≤1
Toughening agent		≤1
Others		6

## Data Availability

Data are contained within the article.

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
