# Peer review of "Fabrication and Polishing Performance of Diamond Self-Sharpening Gel Polishing Disk"

_micromachines, 2023, doi:10.3390/mi15010056_

Round 1

Reviewer 1 Report

Comments and Suggestions for Authors

The authors have proposed a developed a diamond gel polishing disk with self-sharpening ability.  It is very interesting to improve the polishing efficiency and surface quality of 4H-SiC.  While these results and discussions are promising, authors need to revise the manuscript on the following fronts:

1. The authors believe that the sharpening mechanism of the diamond gel polishing disk is the AlN particles with silica 10 sol film coating embedded in the disk matrix are easy to pulled out, resulting in the weak bonding strength of the matrix (Materials Science in Semiconductor Processing, 2022, 143: 106556.  Manufacturing Processes, 2020, 59: 595-603.).

2. In Introduction, the authors should summarize the self-sharpening abrasives and tools, such as agglomerated diamond abrasives (Diamond and Related Materials, 2019, 100: 107595.  Diamond and Related Materials, 2020, 108:107965.)

3. In Fig.17, the authors measure the friction coefficient of polishing disk specimens with different A/S powder 398 content, the friction pair is the silicon nitride ball, the SiC ball maybe suitable.  Moreover, the friction time should be lengthened to 60 min as same as the polishing time.

4. The contact mechanism between the disk and the wafer should be expounded, such as brittle-plastic transition shown in Fig. 14.  Tribology International, 2023, 184: 108493. and Wear, 2021, 464-465:203531. can be as references.

5. In Fig. 9, the AIN/SiO2 particle should be gone through EDS analysis.

Comments on the Quality of English Language

Minor editing of English language required

Reviewer 2 Report

Comments and Suggestions for Authors

Review of the manuscript entitled ‘ Aluminums nitride (AlN) powder with silica 10 sol film coating (A/S powder)’

The authors describe the dissolution property of Aluminums nitride (AlN) powder with silica sol film coating (A/S powder). Friction and wear experiments and polishing experiments were carried out to investigate the effect of A/S powder content on the friction and wear performance of polishing disks and the surface quality of 4H-SiC wafer.

- ‘Figure 14 shows the surface morphology of polishing disk specimens before and after 305 polishing. Figure 12 shows the mechanical properties of three kinds of polishing disks 306 after polishing.’

The figure 14 must be inserted before the figure 12 and renamed.

- ‘the flexural strength and 307 hardness of the polishing disks show an overall decreasing trend as the particle size of the 308 A/S powder increases.’

It seems linear. Is it correct?

- The figure b is not discussed.

- the apparent porosity of the polishing 340 disk specimens is estimated by the drainage method based on Archimedes' principle. Give some words about the method.

- ‘Figure 17 shows the instantaneous coefficient of friction with time for different polishing disks. Figure 15 shows the hardness of the polishing …’

Put the figure 17 before the figure 15. Same thing for figure 20 and figure 19. Reorganize the figures.

- The figure 16 is not discussed.

- The figure 20 is not discussed: ‘Figure 20 shows the surface morphology of 4H-SiC wafers after 407 rough polishing with six types of polishing disks’

What information could you extract from figure 20?

- The figure 22 is not discussed.

Round 2

Reviewer 2 Report

Comments and Suggestions for Authors

The manuscript is now corrected and improved. It can now be accepted.